# Predictability and stability testing to assess clinical decision instrument performance for children after blunt torso trauma

**Aaron E. Kornblith**[1,2ᴼ], **Chandan Singh**[3ᴼ], **Gabriel Devlin**[2], **Newton Addo**[1], **Christian J. Streck**[4], **James F. Holmes**[5], **Nathan Kuppermann**[5,6], **Jacqueline Grupp-Phelan**[1,2], **Jeffrey Fineman**[2], **Atul J. Butte**[7], **Bin Yu**[3,8]*

**1** Department of Emergency Medicine, University of California, San Francisco, San Francisco, United States of America, **2** Department of Pediatrics, University of California, San Francisco, San Francisco, United States of America, **3** Department of Electrical Engineering & Computer Science, University of California, Berkeley, Berkeley, United States of America, **4** Department of Surgery, Medical University of South Carolina, Children's Hospital, Charleston, United States of America, **5** Department of Emergency Medicine, University of California, Davis, Davis, United States of America, **6** Department of Pediatrics, University of California, Davis, Davis, United States of America, **7** Bakar Computational Health Sciences Institute, University of California, San Francisco, San Francisco, United States of America, **8** Departments of Statistics, University of California, Berkeley, Berkeley, United States of America

ᴼ These authors contributed equally to this work.
* binyu@berkeley.edu

**Data Availability Statement:** This work uses the publicly available Pediatric Emergency Care Applied Research Network dataset. Please find access by visiting https://pecarn.org/datasets/. This work also

## Abstract

### Objective

The Pediatric Emergency Care Applied Research Network (PECARN) has developed a clinical-decision instrument (CDI) to identify children at very low risk of intra-abdominal injury. However, the CDI has not been externally validated. We sought to vet the PECARN CDI with the Predictability Computability Stability (PCS) data science framework, potentially increasing its chance of a successful external validation.

### Materials & methods

We performed a secondary analysis of two prospectively collected datasets: PECARN (12,044 children from 20 emergency departments) and an independent external validation dataset from the Pediatric Surgical Research Collaborative (PedSRC; 2,188 children from 14 emergency departments). We used PCS to reanalyze the original PECARN CDI along with new interpretable PCS CDIs developed using the PECARN dataset. External validation was then measured on the PedSRC dataset.

### Results

Three predictor variables (abdominal wall trauma, Glasgow Coma Scale Score <14, and abdominal tenderness) were found to be stable. A CDI using only these three variables would achieve lower sensitivity than the original PECARN CDI with seven variables on internal PECARN validation but achieve the same performance on external PedSRC validation (sensitivity 96.8% and specificity 44%). Using only these variables, we developed a PCS

uses the Pediatric Surgical Research Collaborative dataset. This data is restricted as it may contain potentially identifying or sensitive patient information. For data requests, please contact the Pediatric Surgical Research Collaborative https://www.pedsrc.org. All code is available via Github https://github.com/csinva/iai-clinical-decision-rule.

**Funding:** This work was supported in part by NSF TRIPODS Grant 1740855, DMS-1613002, 1953191, 2015341, IIS 1741340, ONR grant N00014-17-1-2176. Moreover, this work is supported in part by the Center for Science of Information (CSoI), an NSF Science and Technology Center, under grant agreement CCF-0939370. This project was supported in part by the National Center for Advancing Translational Sciences, National Institutes of Health, through UCSF-CTSI Grant Number UL1 TR001872. The funders had no role in study design, data collection and analysis, decision to publish, or preparation of the manuscript.

**Competing interests:** The authors have declared the following competing interests exist. Aaron Kornblith is a co-founder of CaptureDx. Newton Addo is a co-founder of CaptureDx. Atul Butte is a co-founder and consultant to Personalis and NuMedii; consultant to Samsung, Mango Tree Corporation, and in the recent past, 10x Genomics, Helix, Pathway Genomics, and Verinata (Illumina); has served on paid advisory panels or boards for Geisinger Health, Regenstrief Institute, Gerson Lehman Group, AlphaSights, Covance, Novartis, Genentech, and Merck, and Roche; is a shareholder in Personalis and NuMedii; is a minor shareholder in Apple, Facebook, Alphabet (Google), Microsoft, Amazon, Snap, 10x Genomics, Illumina, CVS, Nuna Health, Assay Depot, Vet24seven, Regeneron, Sanofi, Royalty Pharma, AstraZeneca, Moderna, Biogen, Paraxel, and Sutro, and several other non-health related companies and mutual funds; and has received honoraria and travel reimbursement for invited talks from Johnson and Johnson, Roche, Genentech, Pfizer, Merck, Lilly, Takeda, Varian, Mars, Siemens, Optum, Abbott, Celgene, AstraZeneca, AbbVie, Westat, and many academic institutions, medical or disease specific foundations and associations, and health systems. Atul Butte receives royalty payments through Stanford University, for several patents and other disclosures licensed to NuMedii and Personalis. Atul Butte's research has been funded by NIH, Peraton (as the prime on an NIH contract), Genentech, Johnson and Johnson, FDA, Robert Wood Johnson Foundation, Leon Lowenstein Foundation, Intervalien Foundation, Priscilla Chan and Mark Zuckerberg, the Barbara and Gerson

CDI which had a lower sensitivity than the original PECARN CDI on internal PECARN validation but performed the same on external PedSRC validation (sensitivity 96.8% and specificity 44%).

## Conclusion

The PCS data science framework vetted the PECARN CDI and its constituent predictor variables prior to external validation. We found that the 3 stable predictor variables represented all of the PECARN CDI's predictive performance on independent external validation. The PCS framework offers a less resource-intensive method than prospective validation to vet CDIs before external validation. We also found that the PECARN CDI will generalize well to new populations and should be prospectively externally validated. The PCS framework offers a potential strategy to increase the chance of a successful (costly) prospective validation.

## Author summary

Do predictability and stability testing inform how a clinical decision instrument for identifying children at low risk of intra-abdominal injuries undergoing intervention after blunt torso trauma will perform prior to external validation? The PECARN instrument has high prediction performance and stable predictor variables. The Predictability, Computability, Stability (PCS) framework identified high performing instruments after development but before external validation. The PECARN instrument has high predictability and stability for children after blunt torso trauma and should therefore undergo prospective external validation. PCS is an effective method for evaluating clinical decision instruments after development but prior to external validation.

## Introduction

### Background

Blunt intra-abdominal injury is a leading cause of preventable death and disability in children in the U.S [1]. Computed tomography scans (CT) are the reference standard to diagnose intra-abdominal injury. In the last 30 years, CT use in children has increased without proportional improvements in clinical outcomes [2]. Indiscriminate use of CT is associated with an increased risk of radiation-induced malignancy [3]. Uncertainty and the lack of evidence in emergency department risk-stratification strategies lead to wide variation in CT use [4]. Furthermore, variability in practice increases cost and reduces effectiveness, efficiency, and quality of pediatric trauma care [5]. The Pediatric Emergency Care Applied Research Network (PECARN) prospectively developed a clinical decision instrument (CDI) to identify children after blunt torso trauma at very low risk for intra-abdominal injury undergoing acute intervention to decrease indiscriminate CT use [6].

### Importance

Emergency care requires rapid and accurate decisions across a diverse group of patients and practices. CDIs reduce variability for high-prevalence conditions by offering the potential for

Bakar Foundation, and in the recent past, the March of Dimes, Juvenile Diabetes Research Foundation, California Governor's Office of Planning and Research, California Institute for Regenerative Medicine, L'Oreal, and Progenity. Bin Yu has consulted at Microsoft Research, Grail and GenenTech and is a minor shareholder at PatternComputer, Inc. She has received honoraria for invited talks at IQVIA, GenenTech and CapitalOne. She has been funded by CZBiohub with an intercampus research award grant. None of the organizations above participated in the study design, data collection and analysis, decision to publish, or preparation of the manuscript.

more accurate and reliable diagnostic strategies than clinician judgment alone [7]. However, before widespread use, CDIs require external validation. External validation is considered a more robust test of diagnostic performance than internal validation, and is critical to understanding the reliability of CDIs as they are generalized to new populations [8,9]. If the CDI performs poorly during external validation, it may be refined, reconsidered, or even abandoned [10]. However, prospective external validation may be expensive and cumbersome. Therefore, introducing a step to assess a CDI before prospective external validation can ensure that it is developed and modeled to be as predictive and stable as possible, to increase the chance of successful external validation.

Recent progress in data science has led to innovative frameworks to assess the prediction performance and stability of healthcare-related diagnostic models, such as CDIs. The Predictability-Computability-Stability (PCS) framework is a unified approach to data science that protects against instability induced by subjective decisions made during the data science lifecycle [11,12]. PCS has improved drug-response prediction [12], gene-interaction search [13], and drug subgroup discovery in clinical trials [14]; these case-studies suggest that PCS may improve the CDI development and validation process before further investment into external validation. In addition to predictability as a reality check, two critical aspects of PCS are interpretability and stability analysis. To undergo PCS vetting, a CDI must be developed using interpretable methods, ensuring reproducibility [15]. Stability measures how much a CDI varies as choices made during the data science life cycle (including data cleaning and modeling), such as reasonable data alterations or different modeling techniques [16]. Stability is assessed by comparing model-level and variable-level test characteristics to one another.

The specifics of how to measure predictability, computability, and stability are particular to a clinical problem and judgment calls. Here, we show how PCS can be used to assess a CDI at the model-level and variable-level. We show how PCS can be used to assess a CDI at the model-level and variable-level. First, multiple CDIs are developed by subsampling the original PECARN dataset. Second, each CDI is screened at the model-level based on diagnostic test characteristics (predictability) and interpretability. Third, we assess the variability of the importance of different predictor variables across high-performing CDIs (variable-level stability)

## Goals of This Investigation

The primary objective of this study was to demonstrate the use of the PCS data science framework in vetting clinical decision instrument development. The secondary objective was to assess and externally validate the original PECARN clinical decision instrument for identifying children at very low risk of intra-abdominal injuries undergoing acute intervention after blunt torso trauma.

## Results

### Results for Objective 1: Demonstrating the PCS Framework in CDI Development

**Characteristics of study patients.** The PECARN dataset included 12,044 children (Table 1). In PECARN, the mean (SD) age was 10.3 (5.4) years (1,167 patients <2 years), ranging from 0 to 18 years. The PedSRC external validation dataset included 2,188 children. The mean (SD) age was 7.8 (4.6) years (216 patients <2 years), ranging from 0 to 15 years. The PedSRC had a higher prevalence of motor vehicle collisions, compared to the PECARN

**Table 1. Patient demographics and outcomes of the PECARN dataset split into development and validation (80:20), and the PedSRC external validation dataset.**

| | PECARN | | | PedSRC |
|---|---|---|---|---|
| | Total (N = 12,044) | Development (n = 7,985) | Internal Validation (n = 4,059) | External Validation (N = 2,188) |
| Age <2 years (%) | 1167 (9.7%) | 761 (9.5%) | 406 (10%) | 216 (9.9%) |
| Sex Male (%) | 7384 (61.3%) | 4887 (61.2%) | 2497 (61.5%) | N/A |
| MVC (%) | 3832 (31.8%) | 2505 (31.4%) | 1327 (32.7%) | 1014 (46.3%) |
| CT scan (%) | 5,179 (43.0%) | 3,393 (42.5%) | 1,786 (44.0%) | 967 (44.2%) |
| IAI (%) | 761 (6.3%) | 485 (6.1%) | 276 (6.8%) | 261 (11.9%) |
| IAI-I (%) | 203 (1.7%) | 133 (1.7%) | 70 (1.7%) | 62 (2.8%) |

PECARN: Pediatric Emergency Care Applied Research Network; PedSRC: Pediatric Surgery Research Collaborative; MVC: motor vehicle collision; CT scan: computed tomography; IAI: intra-abdominal injury; IAI-I: intra-abdominal injury undergoing acute intervention

development and validation datasets, 46.3% vs. 31.8% and 31.4%, and children with intra-abdominal injuries undergoing acute intervention, 2.8% vs. 1.7% and 1.7%, respectively (Table 1).

## Clinical decision instrument development

We replicated the original PECARN CDI development using the PECARN dataset and re-developed the identical seven ordered decision predictor variables in the PECARN rule list. The potential alternative CDIs, including Bayesian rule lists, CART Decision Trees, CART Rule Lists, Iterative Random Forests, and Rulefit are in S1–S3 Figs and S1 and S2 Tables. The randomness for all re-developed models had no effect on any of the final CDIs performances.

## Clinical decision instrument internal validation

Each CDI had a decline in performance between the development and internal validation PECARN datasets (Fig 1); however, the magnitude of the performance drop differed between different CDIs. The greater the magnitude in reduction suggests a less stable model. For example, the Iterative Random Forest CDI (red) and CART decision tree (orange) had the largest decline in performance between the development and validation datasets, suggesting that the prediction model was overfitting to the development dataset. In contrast, a fitted Bayesian rule

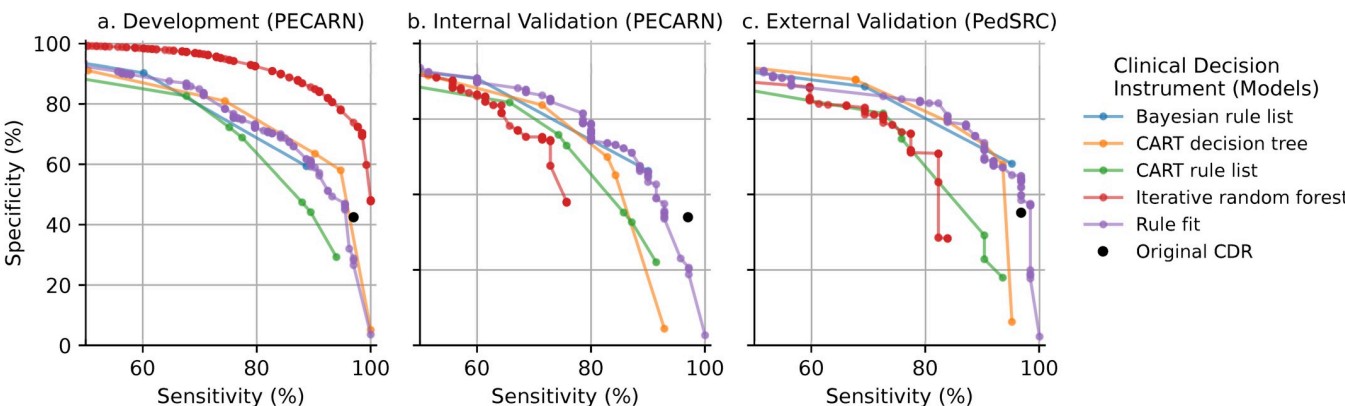

**Fig 1.** Sensitivity-specificity curves for clinical decision instruments to evaluate children after blunt torso trauma on the PECARN (a) development dataset (b) internal validation dataset, and (c) external validation on the PedSRC. The clinical decision instruments were then ranked by predictability from best to worst (top to bottom).

**Table 2. Predictive performance of the clinical decision instruments with sensitivity weighted five times more heavily as specificity.** (a) PECARN Development dataset, (b) PECARN Internal Validation Dataset, (c) PedSRC External Validation Dataset.

| (a) PECARN Development Dataset | PECARN | Bayesian Rule List | CART Decision Tree | CART Rule List | Iterative Random Forest | Rule Fit |
|---|---|---|---|---|---|---|
| Sensitivity | 98% | 89% | 95% | 94% | 98% | 95% |
| Specificity | 43% | 59% | 58% | 29% | 70% | 47% |
| Negative predictive value | 99.9% | 100% | 100% | 100% | 100% | 100% |
| Positive predictive value | 2.8% | 4% | 4% | 2% | 5% | 3% |
| Negative likelihood ratio | 0.035 | 0.19 | 0.09 | 0.21 | 0.02 | 0.1 |
| Positive likelihood ratio | 1.74 | 2.18 | 2.26 | 1.33 | 3.33 | 1.80 |
| F1 score | 0.056 | 0.07 | 0.07 | 0.04 | 0.10 | 0.06 |
| Brier score | 0.016 | 0.02 | 0.58 | 0.02 | 0.01 | 0.08 |
| (b) PECARN Internal Validation Dataset | PECARN | Bayesian Rule List | CART Decision Tree | CART Rule List | Iterative Random Forest | Rule Fit |
| Sensitivity | 94% | 90% | 84% | 91% | 71% | 97% |
| Specificity | 41% | 58% | 56% | 28% | 68% | 33% |
| Negative predictive value | 99.8% | 100% | 100% | 99% | 99% | 100% |
| Positive predictive value | 2.7% | 4% | 3% | 2% | 4% | 2% |
| Negative likelihood ratio | 0.14 | 0.17 | 0.28 | 0.31 | 0.42 | 0.09 |
| Positive likelihood ratio | 1.60 | 2.13 | 1.93 | 1.26 | 2.24 | 1.45 |
| F1 score | 0.053 | 0.07 | 0.06 | 0.04 | 0.07 | 0.04 |
| Brier score | 0.016 | 0.02 | 0.59 | 0.02 | 0.02 | 0.08 |

| (c) PedSRC External Validation Dataset | PECARN | Bayesian Rule List | CART Decision Tree | CART Rule List | Iterative Random Forest | Rule Fit |
|---|---|---|---|---|---|---|
| Sensitivity | 96.8% | 95% | 94% | 90% | 81% | 97% |
| Specificity | 44.0% | 60% | 60% | 39% | 63% | 55% |
| Negative predictive value | 99.8% | 100% | 100% | 99% | 99% | 100% |
| Positive predictive value | 4.8% | 7% | 6% | 4% | 6% | 6% |
| Negative likelihood ratio | 0.073 | 0.08 | 0.11 | 0.25 | 0.30 | 0.06 |
| Positive likelihood ratio | 1.73 | 2.39 | 2.33 | 1.47 | 2.21 | 2.13 |
| F1 score | 0.091 | 0.12 | 0.12 | 0.07 | 0.11 | 0.11 |
| Brier score | 0.026 | 0.03 | 0.56 | 0.03 | 0.03 | 0.09 |

list (blue), CART rule list (green), and Rule fit (purple) all retained similar predictive accuracy between development and validation. Table 2 summarizes the results of threshold-specific weights in which the sensitivity is weighted five times more heavily as specificity.

## Predictability

The original PECARN, Rule Fit, and Bayesian Rule List had minimal changes in performance between the development and internal validation datasets, suggesting relatively high predictability for these CDIs (Fig 1B). In contrast, CART Rule List, CART Decision Tree, and Iterative Random Forest had greater proportional declines in performance, suggesting lower predictability when heterogeneity in datasets was introduced.

## Predictor-variable stability

The most stable predictor variables were *abdominal trauma/seat belt sign*, *Glasgow Coma Scale Score < 14*, and *abdominal tenderness*. These three variables were the most frequent recurring predictor variables between CDIs. These three variables also had the highest non-zero permutation scores between the different CDIs (S4 Fig). Therefore, it was recognized that the top

three performing predictor variables were selected in the PECARN CDI and the four top-performing CDIs.

## Computability

Computability assesses the computational needs (e.g., hardware and demand for specialized equipment) of the project to understand the efficiency and feasibility of repeating the task. We evaluated the computational needs for CDI development and validation by timing each epoch and run time. In this case, all modeling and data analysis was performed on a standard laptop computer: CDI development took less than 10 minutes and validation less than 1 second.

## Results for objective 2: External validation of original PECARN clinical decision instrument

**Distributions and variable matching for the external validation dataset.** Predictor and outcome variables between the PECARN and PedSRC datasets were matched and evaluated for variable-level distributions (Fig 2). Most variables had direct matches between datasets (S3–S5 Tables). Predictor variables were assessed for redundancy and independent associations (S5 Fig). The distribution of variables was well-matched except for the PECARN dataset inclusion of patients 15–17 years, and the lower frequency of children presenting after motor vehicle collisions (MVC) (Fig 2).

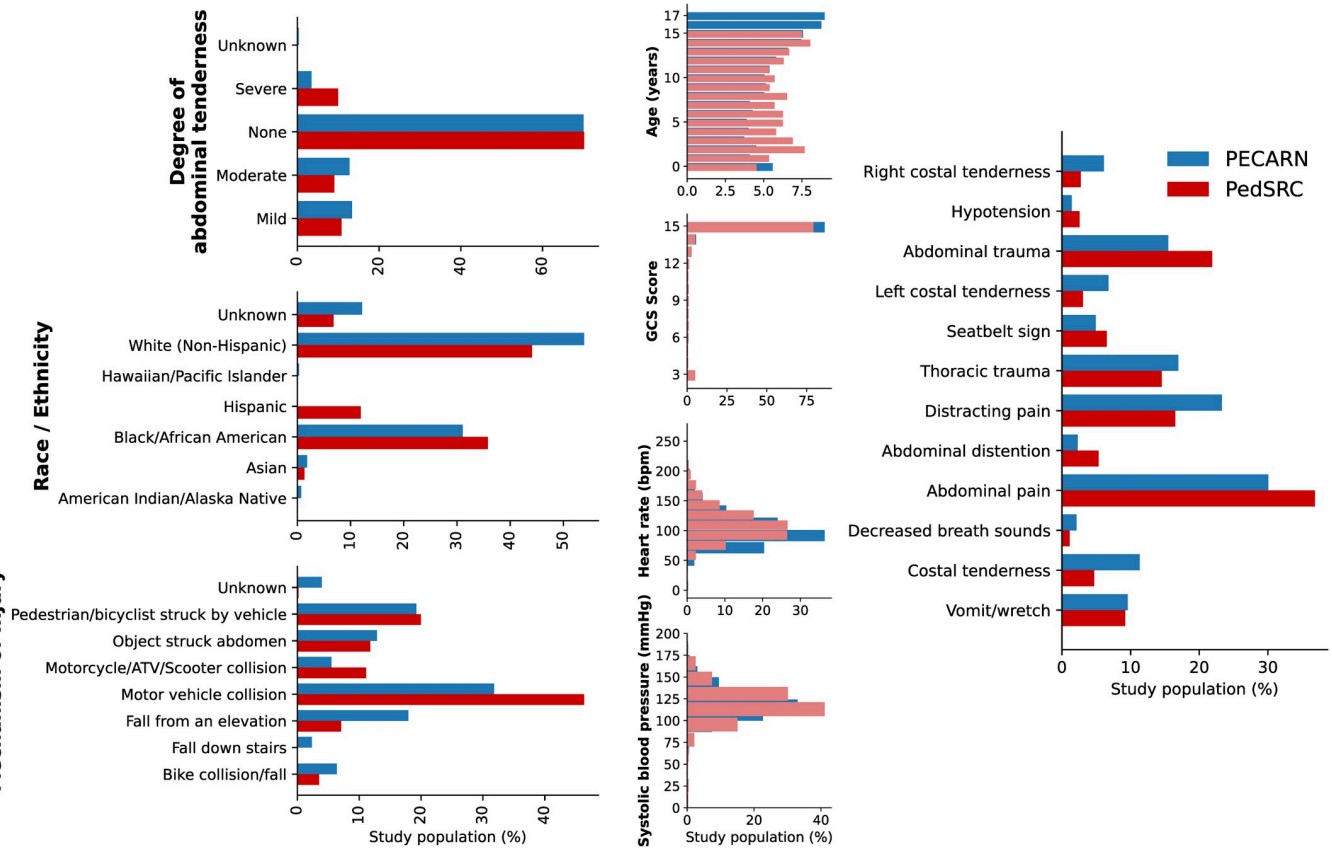

**Fig 2. Matched demographic and predictor variables from PECARN and PedSRC datasets visually represented for overall distributions.**

### External validation predictive performance

The original PECARN CDI successfully identified all but six children with intra-abdominal injuries undergoing acute interventions (sensitivity 97%, specificity 42.5%) on the PECARN dataset (Table 2B). On external validation using the PedSRC dataset, the original PECARN CDI maintained high prediction performance with an external validation sensitivity of 97.0% and specificity 44.0% (Table 2C). However, the original PECARN CDI missed two children with intra-abdominal injuries undergoing acute interventions; the clinical characteristics of these two children are presented in S6 Table.

The three predictor variables that were found to be stable on PCS testing identified 60/62 patients with intra-abdominal injuries undergoing acute interventions in the PedSRC dataset, corresponding to 100% of the PECARN CDI's predictive power (blue box, Fig 3). The remaining four predictor variables (red box, Fig 3) did not add to the predictive performance of the PECARN CDI on the PedSRC dataset. The Brier score was 0.026, suggesting the predicted risk is well-calibrated when using the original PECARN CDI. The same three predictor variables also captured the majority of the predictive power in the original PECARN validation,

**Fig 3.** Prediction tree for the original PECARN clinical decision instrument on (a) PECARN internal validation dataset, and (b) PedSRC external validation dataset. The blue box shows that the top three predictor variables retained all the predictive power for the clinical decision instrument on external validation. The red box shows the predictor variables without prediction power on external validation. From the top of the rule to the bottom, risks for the identified subgroups monotonically decrease, although risks are systematically higher on the PedSRC data.

identifying 186 of 201 outcomes. The additional four predictor variables substantially reduce the CDI's specificity. However, without these variables, the CDI misses 11 IAI-I patients in the PECARN dataset, resulting in unacceptably low sensitivity.

## Objective 1 & objective 2: Comparing pcs framework predictions to external validation of clinical decision instruments

The ranked overall performance of the CDI on external validation matched that of the PCS framework prediction rankings (Fig 1C). This suggests that the results obtained from the PCS framework yielded useful information about the CDI's external validation performance, prior to collecting or analyzing the external validation dataset. In addition, the predictive performance was similar between internal validation and external validation (using the PedSRC dataset). However, most CDIs slightly improved their performances, suggesting that the CDIs are not overfitting to the PECARN dataset (Table 2C). The original PECARN CDI, Bayesian rule list, and Rule Fit had similar performances as in the PECARN datasets. In contrast, Iterative Random Forest, CART decision tree, and CART rule list had large declines in predictive performance (Fig 1C).

## Discussion

In the discussion, first we seek to describe PCS in the context of CDI development and vetting focusing on three key topics: predictability, stability, and interpretability. Next, we exemplify these three topics and their implications for the PECARN CDI.

### Contextualizing PCS in the context of CDI development

**Predictability.**   The predictive performance of a CDI serves as the benchmark in the clinical literature. The concept of diagnostic test characteristics, such as sensitivity and specificity, are well-described and clinically used metrics for predictability. For example, previous literature has found that the PECARN CDI has a higher sensitivity than clinical judgment alone [17]. This study sought to evaluate the predictability of a CDI using threshold-dependent discriminative metrics (i.e., sensitivity) and threshold-free metrics (i.e. sensitivity-specificity curves). We found that the PECARN, Bayesian, and Rule Fit CDIs were the most predictable on external validation (PedSRC). However, CDIs used in clinical practice are designed to make predictions on varying populations, over time, and within differing conditions. Therefore, before using a CDI in clinical practice, investigators should validate how well a CDI will perform under varying conditions.

**Stability.**   Stability should be checked for all aspects of the data science lifecycle. Here, we largely focus on predictor-level stability, estimating how the feature importance of each predictor variable changes as a result of different judgment calls made during modeling. We also examine the stability of both the predictive performance and individual predictors to different calls made during data preprocessing. For example, we tried using GCS as a continuous predictor variable compared to different binary thresholds. The effect of this and many other judgment calls were found to be minimal and are omitted here (but can be found on our github).

**Interpretability.**   Interpretability enables the integration of domain expertise for the development and implementation of a CDI [18–20]. In contrast, black-box machine-learning models lack interpretability and may fail for unknown reasons when externally validated [21]. Post-hoc interpretations, such as permutation importance used here, can offer some interpretability [22–25], but are not a substitute for developing an interpretable model [15,26–28]. Therefore, we only consider parsimonious rule-based models. Each CDI is represented as a

straightforward set or list of logical rules (IF:THEN statements), which can then be visualized. We restrict each model to a reasonable number of logical steps (fewer than 10), so each CDI can be assessed in real-time. We additionally fit logistic regression and optimal decision tree models, but found that they had poor; we find that fast interpretable greedy-tree sums learn precisely the same rules as CART so we omit this model here. PCS offers clear documentation guidelines to ensure the process is replicable, reproducible, and interpretable [11].

As stated, black-box machine-learning models lack interpretability and may fail for unknown reasons when tested on new populations [19]. Examples of such complex models are neural networks, random forests, and support vector machines. However, even seemingly simple models such as logistic regression or decision trees can become uninterpretable if they are large enough and have too many steps [15]. Pennell (2020) utilized such models to re-evaluate the PECARN dataset [29]. The authors concluded that they had developed and validated a novel risk model using modern machine learning techniques. However, these complex machine-learning models lack the interpretability to integrate judgment, thus not allowing review nor the recognition of bias, which may build mistrust in the user [20]. Therefore, we use interpretable models with visual representation to allow stability analysis and ensure the integration of clinical judgment within the CDI [18].

## Implications for the PECARN CDI

As the second aim of this paper, we assessed the prediction performance and the stability of the original PECARN CDI for identifying children at very low risk of intra-abdominal injuries undergoing acute intervention after blunt torso trauma on external validation. Clinically, there is no standard, generalizable, validated strategy to identify children after blunt torso trauma in whom CT scans can safely be avoided. Instead, providers use ad hoc strategies that are inaccurate, and may fail to identify life-threatening injuries, leading to over-reliance on diagnostic imaging [30–33]. In 2013, PECARN sought to address the variability in accuracy and consistency by prospectively developing a CDI for children after blunt torso trauma [6].

We used two uniquely matched prospectively collected but independent datasets to assess the CDI predictions and stability on external validation. Through this process, we reexamined the original PECARN findings using alternative reasonable statistical models and found the original PECARN CDI to be high performing. The PECARN CDI was highly predictive across the development, internal validation, and external validation datasets. Therefore, PECARN has strong predictive performance, which measures how well a CDI predicts in heterogeneous cohorts. We also found that three predictor variables made up the entirety of the predictive power on external validation: abdominal wall trauma, Glasgow Coma Scale Score <14, and abdominal tenderness. This is not surprising, as these three variables were also the most stable based on the PCS framework and made up the majority of the predictive power on the PECARN dataset (identifying 94.4% of the correctly predicted IAI-I patients).

Through the PCS framework, we found that the predictability, and stability of the original PECARN CDI warrants further investment and investigation, including prospective external validation. In contrast, if we found that the model or predictor variables were unstable in the original study, we would recommend against further validation. Our study can serve as an example for how investigators may evaluate the predictability and stability of a CDI for inherent weakness, prior to investing in a prospective external validation.

We found that if PCS could be successfully integrated as a novel step into prediction and diagnostic model development before external validation, there is a potential to streamline and evaluate CDIs to improve performance or expose weaknesses and avoid further investment in CDIs with poor stability. This is important because many CDIs have reduced accuracy during

external validation [34]. Introducing a PCS step between CDI development and external validation, or using PCS directly for CDI development before external validation, will allow researchers, funders, and clinicians to understand better how CDIs may perform on future populations before external validation, impact analysis, or implementation into clinical practice. However, PCS is not able to replace external validation.

There are limitations to this study. First, we sought to develop high performing but interpretable CDIs. Therefore, we chose only rule-based models, including simple regression-based and complex machine learning models with interpretable visual outputs. The inclusion of less interpretable models may have improved diagnostic accuracy but interfered with conducting stability analysis, introducing domain expertise, and more easily recognizing bias. Second, the PECARN and PedSRC datasets were collected from different research groups. There is a potential for partial verification bias on external validation because the PedSRC dataset was not based on consecutive patient enrollment, and follow-up was limited to medical record review. Third, three predictor variables did not match between datasets. Two variables could not be matched because they were present in only one of the datasets: *gender* (PECARN only) and *femur fracture* (PedSRC only). The third predictor variable was *distracting injury* (prospectively collected in PECARN but retrospectively aggregated in PedSRC). Given the limitations of this study, we believe prospective external validation is required before implementing the CDI.

In conclusion, the PCS data science framework helped vet CDI predictive performance and stability before external validation. The PCS framework offers a computational and less resource-intensive method than external validation. Even though it does not replace prospective external validation, PCS offers a method to vet for unstable CDIs to avoid further investment. We found that the predictive performance and stability of the PECARN CDI warranted further investigation, including prospective external validation. We used the external PSRC dataset to carry out this investigation, validating the PECARN CDI and a similar but simpler PCS-driven CDI.

## Methods

### Ethics statement

This study is a secondary analysis of two datasets. The study protocols are described in the original trial investigations, PECARN [6] and PedSRC [35]. The institutional review boards at each participating site approved the original studies. This secondary analysis of anonymized data was deemed exempt from review by the University of California, San Francisco, and Medical University of South Carolina institutional review boards. Confidentiality was maintained by using only de-identified data. All analysis was performed on de-identified data without access to a lookup key. All authors completed Human Subjects research requirements.

We analyzed two independent prospectively collected datasets from two large pediatric research networks, PECARN and the Pediatric Surgical Research Collaborative (PedSRC). There were two objectives of this study. The first (Section 2.1) was to demonstrate the PCS framework for improving CDI development. The second (Section 2.2) was to assess prediction performance and stability of the original PECARN CDI on external validation.

### Objective 1: Demonstrate predictability-computability-stability (PCS) data science framework for improving cdi development

We followed the PCS framework, which goes beyond traditional reporting guidelines to assess the impact of reasonable human judgment calls by conducting reasonable data/model

perturbations across the entire data science lifecycle [7,16]. PCS offers a framework to assess a CDI for diagnostic performance based on predictive performance (i.e. sensitivity and specificity) and computational needs, putting weight on stability. During the development of a CDI, investigators make many "judgment calls", i.e. subjective decisions which may lead to variability in the final developed CDI. PCS recommends that investigators ensure that study conclusions are stable to any such judgment calls. These judgment calls can be checked by measuring the stability of conclusions when alternative "reasonable" judgment calls are made. Reasonable judgment calls are those solicited through direct engagement between clinicians and data scientists (see the Discussion section for a more detailed look at PCS in the context of CDIs).

In this study, the PCS framework was applied to CDI development (S6 Fig), including all CDI development and validation stages (it could also be applied to the data cleaning stage, but was not done here). First, the PCS framework (1) defines the clinical problem, then reviews all aspects of (2) collecting and preprocessing data, and (3) develops CDIs using interpretable and rule-based models. Next, these CDIs are vetted for their (4) predictive performance (predictability) and the importance of predictor variables. Last, PCS (5) supports the interpretation of results by identifying variability in all the PCS steps (stability), ensuring CDIs are developed to be supported by both data and domain knowledge (provider input). In addition, PCS guided all aspects of data documentation and analysis; code is available on Github (https://github.com/csinva/iai-clinical-decision-rule) [10].

**Development and validation dataset.** The PECARN dataset is a prospective cohort of 12,044 children after blunt torso trauma between May 2007 and January 2010 in 20 emergency departments [6]. Predictor variables were collected prospectively using a standard data collection tool. We used the PECARN definition for the a priori outcome of interest of intra-abdominal injury undergoing acute intervention [6].

Following the original PECARN methods, we excluded any variable that was missing more than 5%, and used predictor variables with at least moderate inter-rater agreement, with the lower bound of the 95% confidence interval (CI) of the k measurements being at least 0.4 [17]. Missing values for a predictor variable were imputed via its median, and we manually combined predictors that conveyed redundant information based on their correlations (S5 Fig).

**Original PECARN CDI development.** Redevelopment of the PECARN CDI ensures the replicability of the original trial. We followed the original PECARN development and internal cross-validation process to redevelop the PECARN CDI to identify children at very low risk for intra-abdominal injuries undergoing acute intervention [6]. We used a Classification and Regression Trees (CART) rule list [36], which involves binary recursive partitioning using the Gini criterion [37].

**PCS CDI development.** We developed several alternative CDIs (corresponding to different judgment calls during modeling) to compare the predictive performance and perform stability analysis of the PECARN CDI. A critical difference of PCS and the original PECARN study is that we use a sample-split, which ensures that our test characteristics are more reliable and do not overestimate a CDI's performance due to overfitting. The following models were used to develop CDIs: logistic regression, CART decision trees, rule lists, [36] Bayesian Rule Lists [38], iterative Random Forests [13], RuleFit [39], Optimal sparse decision trees [40], Fast interpretable greedy-tree sums [26] and manual subgroup analysis. Each rule-based predictive model was chosen for its interpretability, taking the form of either a parsimonious list, tree, or set of binary rules. We used a stratified splitting technique to divide the PECARN dataset into a development set (i.e. a training set), 7,985 children (66%), and a validation set, 4,059 children (34%). Predictive models were fit using the imodels python package [27] (version 0.2.5). Hyperparameters were selected via manual tuning using only the development dataset.

**CDI predictive performance.** We calculated standard diagnostic statistics to report CDI performance. We used sensitivity and specificity curves to compare the diagnostic test characteristics of each CDI in the PECARN development and internal validation datasets. Furthermore, many more test characteristics were reported for each CDI, including their positive predictive value and Brier score (which helps evaluate the calibration of a CDI) [41]. The CDIs were ranked heuristically from the sensitivity-specificity curves by weighting (threshold-dependent) sensitivity five times more than specificity. CDIs with poor predictive performance (i.e., achieving a sensitivity below 90%) were eliminated before further analysis.

**CDI stability.** We assessed CDI stability by performing side-by-side comparisons of the PECARN CDI and alternative CDIs. The original PECARN study does not report stability at the model-level or variable level. To assess predictor-variable stability, we report the frequency and non-zero permutation-importance score of each predictor variable for each CDI [42]. The permutation importance measures the effect a predictor variable has on the overall prediction model's error. If a predictor variable is important, permuting or shuffling the value increases the model's error. The predictor variables with high permutation importance, especially across many different CDIs have greater stability.

We also compared the variability of diagnostic test characteristics between the PECARN development and internal validation datasets to assess the generalization of the model (i.e. stability of the predictive performance). Large changes in test characteristics suggest that the model is unstable in generalizing to new data. Moreover, a CDI can be unstable even when being re-developed to the same data. This is because many models contain some randomness in fitting, which can produce a different result when a model is re-developed. Therefore, we also measure randomness when each model is re-developed as a marker of stability. Prediction models were then ranked based on predictive performance (sensitivity and specificity), and then on variable-level stability.

## Objective 2: Predictability and stability of the original PECARN CDI on external validation

**External validation dataset.** The PedSRC dataset is based on a prospective cohort of 2,188 children with blunt trauma at 14 non-PECARN Level I pediatric trauma centers [35]. Predictor variables were collected prospectively using a standard data collection tool. The PedSRC study defined intra-abdominal injury as any injury to an intra-abdominal structure identified on abdominal CT or at laparotomy. We matched the a priori PedSRC outcome of intra-abdominal injury undergoing intervention to the PECARN outcome.

We matched predictor and outcome variables between the datasets through distribution assessment and expert review. To ensure consistent matching, all variable linkages between datasets were reviewed by domain experts, including PECARN and PedSRC study principal investigators, to ensure biologic plausibility and ensure original data definition was congruent between the respective datasets. Variables with subjectivity were further screened, original documentation reviewed, and expert authorship team consensus was used to match variables. The same missing data strategy was used on the PedSRC and PECARN datasets.

**PCS external validation.** To externally validate each CDI, we calculated threshold-bound and threshold-free standard diagnostic statistics. We calculated sensitivity, specificity, negative and positive predictive values, positive and negative likelihood ratios. We also included false positives, false negatives, accuracy, and F1 score. The F1 score, an accuracy indicator, emphasizes the clinical relevance of sensitivity over specificity and ranges from 1 (best value) to 0 (worst value). We used sensitivity-specificity curves to compare the test characteristics of each candidate CDI on the external test dataset. We ranked predictor variable importance by

assessing each variable's redundancy and weighted predictive power on the external validation dataset. Finally, we assessed overall CDI performance by evaluating the diagnostics test characteristics and variable importance.

We considered clinical context, predictive performance, computational speed, and stability to assign each CDI a rank. To compare the PCS framework to external validation, we first ranked predictive performance and stability. As the goal of the CDI is to limit unnecessary CT use in children after blunt torso trauma, we set a comparison threshold for predictive performance as a sensitivity five times more than specificity with a lower bound sensitivity of at least 95%. We calculated standard diagnostic statistics to report CDI performance, including sensitivity, specificity, negative and positive predictive values, positive, negative likelihood ratios, false positives, false negatives, accuracy, and F1 score. We also ranked predictor variable importance by assessing each variable's redundancy and weighted predictive power on the external validation dataset. Similarly, we measured overall stability as the proportion of the CDI's predictive performance assigned to predictor variables with the highest and lowest variable-level stability.

## Supporting information

**S1 Table. Iterative Random Forest permutation importance scores.**
(DOCX)

**S2 Table. RuleFit.**
(DOCX)

**S3 Table. Predictor variables with 1:1 match between the two study datasets, PECARN and PedSRC.**
(DOCX)

**S4 Table. Predictor variables that were mapped between the two study datasets, PECARN and PedSRC.**
(DOCX)

**S5 Table. Predictor variables that were adjudicated or left out between the two study datasets, PECARN and PedSRC.**
(DOCX)

**S6 Table. Children with intra-abdominal injury requiring acute intervention predicted very low risk by the original PECARN clinical decision instrument on the PedSRC external validation dataset.**
(DOCX)

**S1 Fig. Bayesian rule list.**
(EPS)

**S2 Fig. CART decision tree.**
(EPS)

**S3 Fig. CART rule list.**
(EPS)

**S4 Fig. Non-zero permutation importance scores for predictor variables across high-performing clinical decision instruments.**
(EPS)

**S5 Fig.** Redundant predictor variables are compared in this heatmap of (a) all predictors and (b) subset of key predictors. Darker blue signifies a direct correlation. Darker red signifies an inverse correlation. White signifies no correlation.
(EPS)

**S6 Fig. Five stages of the Predictability Computability Stability (PCS) framework as adapted for clinical decision instrument assessment.** First, the PCS framework (1) defines the clinical problem, then reviews all aspects of (2) collecting and preprocessing data, and (3) models clinical decision instruments using interpretable and rule-based models. Next, the PCS framework (4) performs a validity (predictability), stability, and validation analysis. Last, the PCS framework (5) supports the interpretation of the results by identifying limitations in all the PCS steps, ensuring clinical decision instruments are developed to be supported by both data and domain expertise.
(EPS)

## Author Contributions

**Conceptualization:** Aaron E. Kornblith, Chandan Singh, Christian J. Streck, Bin Yu.

**Data curation:** Aaron E. Kornblith, Chandan Singh, James F. Holmes, Nathan Kuppermann, Bin Yu.

**Formal analysis:** Aaron E. Kornblith, Chandan Singh, Newton Addo, Christian J. Streck, Bin Yu.

**Funding acquisition:** Aaron E. Kornblith, Bin Yu.

**Methodology:** Aaron E. Kornblith, Bin Yu.

**Supervision:** Aaron E. Kornblith, Chandan Singh, Christian J. Streck, James F. Holmes, Nathan Kuppermann, Bin Yu.

**Writing – original draft:** Aaron E. Kornblith, Chandan Singh, Christian J. Streck, Bin Yu.

**Writing – review & editing:** Aaron E. Kornblith, Chandan Singh, Gabriel Devlin, Newton Addo, Christian J. Streck, James F. Holmes, Nathan Kuppermann, Jacqueline Grupp-Phelan, Jeffrey Fineman, Atul J. Butte, Bin Yu.

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
