## [Decision Letter · Decision Letter 0]

24 May 2022

PDIG-D-22-00077

Predictability and Stability Testing to Assess Clinical Decision Instrument Performance for Children After Blunt Torso Trauma

PLOS Digital Health

Dear Dr. Yu,

Thank you for submitting your manuscript to PLOS Digital Health. After careful consideration, we feel that it has merit but does not fully meet PLOS Digital Health's publication criteria as it currently stands. Therefore, we invite you to submit a revised version of the manuscript that addresses the points raised during the review process.

As noted, reviewers have some questions about the validity, utility and future prospects of the proposed PCS CDI framework. 

Please submit your revised manuscript by . If you will need more time than this to complete your revisions, please reply to this message or contact the journal office at digitalhealth@plos.org. Please include the following items when submitting your revised manuscript:

We look forward to receiving your revised manuscript.

Kind regards,

Nicole Yee-Key Li-Jessen

Academic Editor

PLOS Digital Health

Journal Requirements:

State what role the funders took in the study. If the funders had no role in your study, please state: “The funders had no role in study design, data collection and analysis, decision to publish, or preparation of the manuscript.”

2. Please update your Competing Interests statement. If you have no competing interests to declare, please state: “The authors have declared that no competing interests exist.”

3. Please provide separate figure files in .tif or .eps format only and remove any ensure that all files are under our size limit of 10MB.

For more information about how to convert your figure files please see our guidelines: https://journals.plos.org/digitalhealth/s/figures

Additional Editor Comments (if provided):

Reviewers' comments:

Reviewer's Responses to Questions

**Comments to the Author**

1. Does this manuscript meet PLOS Digital Health’s publication criteria? Is the manuscript technically sound, and do the data support the conclusions? The manuscript must describe methodologically and ethically rigorous research with conclusions that are appropriately drawn based on the data presented.

Reviewer #1: Yes

Reviewer #2: Yes

Reviewer #3: Yes

2. Has the statistical analysis been performed appropriately and rigorously?

Reviewer #1: I don't know

Reviewer #2: Yes

Reviewer #3: I don't know

3. Have the authors made all data underlying the findings in their manuscript fully available (please refer to the Data Availability Statement at the start of the manuscript PDF file)?

Reviewer #1: Yes

Reviewer #2: Yes

Reviewer #3: Yes

4. Is the manuscript presented in an intelligible fashion and written in standard English?

Reviewer #1: Yes

Reviewer #2: Yes

Reviewer #3: Yes

5. Review Comments to the Author

Reviewer #1: Thank you for submitting this piece. 

As I understand it, the aim was to use a novel framework for data science to evaluate a current CDI. The reason for doing so is to evaluate the likelihood that the CDI would be successful when implemented prospectively. It would seem that the opposite scenario would be failure of a CDI based on the PCS framework would mean some additional work should be done before implementing a CDI.

This paper has many robust assessments of the PECARN and PedSRC data sets. A question that seemed to be raised was, what measures of Predictability, Computability, and Stability are or have been used in the past? It seemed as though algorithmic quality indicators were being used to evaluate a tool perhaps on different axes? For example, while specificity and sensitivity are used in Figure 1, the blunt trauma CDI is represented by a single point, which is slightly difficult to compare characteristically with the other curves. 

Similarly, it was not clear how or why the elements in the CDS framework are chosen and how best to calculate them. It might be that there is no specific approach to measuring these qualities, but if so, perhaps it could be highlighted in the introduction. The introduction introduced the PCS framework, but not necessarily how it was used to identify or improve other models. Did the act of assessing the models in itself lead to further investigation of important variables, or does the appropriate use of the PCS framework lead to results which can be directly employed in the model? It does not seem so, but it was not clear what to expect from the use of PCS on this CDI.

Overall, this was a very robust and well-written manuscript, but it might benefit from some more direct language in the introduction and methods which might then make the results more easy to interpret. It seemed as though the results themselves could not be used to directly determine if the CDI "passed" a PCS evaluation as it simply gave numbers which did not have any context with which to evaluate how good or poor the performance was or what was limiting that performance. 

Just some general questions regarding results were that you mention you then developed a PCS CDI from three variables. Is this a necessary approach to using PCS framework or an additional, perhaps third aim? It was not clear that developing a new CDI was a direct result of the approach mentioned in the methods.

Reviewer #2: This was a very well written and informative article evaluating the PECARN CDI for blunt abdominal trauma in children using the predictability computability stability (PCS) data science framework. The clinical problem the authors seek to address is very relevant. The methodology and the results are presented nicely and conclusions is valid.

Reviewer #3: Excellent Manuscript that will establish process for future studies. Thank you for completing this and submitting. The discussion does have a section detailing what the impact is on the PECARN CDI, and if I am reading correctly it will now proceed to external validation, without additional changes or study? While this is discussed, I am left a little uncertain as to exactly what are the planned next steps - and I am sure that the readers will be looking for that clarity in the manuscript.

6. PLOS authors have the option to publish the peer review history of their article (what does this mean?). If published, this will include your full peer review and any attached files.

**Do you want your identity to be public for this peer review?** For information about this choice, including consent withdrawal, please see our Privacy Policy.

Reviewer #1: No

Reviewer #2: No

Reviewer #3: No

---

## [Decision Letter · Decision Letter 1]

14 Jun 2022

Predictability and Stability Testing to Assess Clinical Decision Instrument Performance for Children After Blunt Torso Trauma

PDIG-D-22-00077R1

Dear Professor Yu,

We are pleased to inform you that your manuscript 'Predictability and Stability Testing to Assess Clinical Decision Instrument Performance for Children After Blunt Torso Trauma' has been provisionally accepted for publication in PLOS Digital Health.

Best regards,

Nicole Yee-Key Li-Jessen

Academic Editor

PLOS Digital Health

Reviewer Comments (if any, and for reference):

Reviewer's Responses to Questions

**Comments to the Author**

1. If the authors have adequately addressed your comments raised in a previous round of review and you feel that this manuscript is now acceptable for publication, you may indicate that here to bypass the “Comments to the Author” section, enter your conflict of interest statement in the “Confidential to Editor” section, and submit your "Accept" recommendation.

Reviewer #2: All comments have been addressed

Reviewer #3: All comments have been addressed

2. Does this manuscript meet PLOS Digital Health’s publication criteria? Is the manuscript technically sound, and do the data support the conclusions? The manuscript must describe methodologically and ethically rigorous research with conclusions that are appropriately drawn based on the data presented.

Reviewer #2: Yes

Reviewer #3: Yes

3. Has the statistical analysis been performed appropriately and rigorously?

Reviewer #2: Yes

Reviewer #3: Yes

4. Have the authors made all data underlying the findings in their manuscript fully available (please refer to the Data Availability Statement at the start of the manuscript PDF file)?

Reviewer #2: Yes

Reviewer #3: Yes

5. Is the manuscript presented in an intelligible fashion and written in standard English?

Reviewer #2: Yes

Reviewer #3: Yes

6. Review Comments to the Author

Reviewer #2: I thank the authors for their responses to the reviewers questions.

Reviewer #3: Prior Comments addressed very well. There is improved clarity on the impact and next steps for the original project.

7. PLOS authors have the option to publish the peer review history of their article (what does this mean?). If published, this will include your full peer review and any attached files.

**Do you want your identity to be public for this peer review?** For information about this choice, including consent withdrawal, please see our Privacy Policy.

Reviewer #2: No

Reviewer #3: No
